# Analysis of Excess All-Cause Mortality and COVID-19 Mortality in Peru: Observational Study

**DOI:** 10.3390/tropicalmed7030044

**Published:** 2022-03-05

**Authors:** Max Carlos Ramírez-Soto, Gutia Ortega-Cáceres

**Affiliations:** 1Facultad de Ciencias de la Salud, Universidad Tecnológica del Perú, Lima 15046, Peru; 2Escuela de Posgrado, Universidad Ricardo Palma, Lima 15039, Peru; gutiaortega@gmail.com

**Keywords:** excess mortality, COVID-19, mortality, Peru

## Abstract

During the COVID-19 pandemic, an excess of all-cause mortality has been recorded in several countries, including Peru. Most excess deaths were likely attributable to COVID-19. In this study, we compared the excess all-cause mortality and COVID-19 mortality in 25 Peruvian regions to determine whether most of the excess deaths in 2020 were attributable to COVID-19. Excess deaths were calculated as the difference between the number of observed deaths from all causes during the COVID-19 pandemic (in 2020) and the number of expected deaths in 2020 based on a historical from recent years (2017–2019). Death data were retrieved from the Sistema Informatico Nacional de Defunciones (SINADEF) at the Ministry of Health of Peru from January 2017 to December 2020. Population counts were obtained from projections from Peru’s Instituto Nacional de Estadística e Informática (INEI). All-cause excess mortality and COVID-19 mortality were calculated by region per 100,000 population. Spearman’s test and linear and multiple regression models were used to estimate the correlation between excess all-cause mortality and COVID-19 mortality per 100,000 population. Excess all-cause death rates varied widely among regions (range: 115.1 to 519.8 per 100,000 population), and COVID-19 mortality ranged between 83.8 and 464.6 per 100,000 population. There was a correlation between the all-cause excess mortality and COVID-19 mortality (r = 0.90; *p* = 0.00001; y = 0.8729x + 90.808; R^2^ = 0.84). Adjusted for confounding factors (mean age in the region, gender balance, and number of intensive care unit (ICU) beds), the all-cause excess mortality rate was correlated with COVID-19 mortality rate (β = 0.921; *p* = 0.0001). These findings suggest that most of the excess deaths in Peru are related to COVID-19. Therefore, these findings can help decision-makers to understand the high COVID-19 mortality rates in Peru.

## 1. Introduction

During the COVID-19 pandemic, an all-cause mortality excess has been recorded in several countries, including Peru [1,2,3,4]. This all-cause mortality excess varied substantially across countries [1,2,3] because of measures taken to handle the COVID-19 pandemic, demographic and socio-economic characteristics, and capacity of health care systems [5,6,7,8]. Excess deaths are the difference between the number of observed deaths from all causes during a given time period, and the number of expected deaths from the same time period, based on a historical from recent years (often estimated using the average over several preceding years) [8]. Excess all-cause mortality can also be standardized for age, sex, region, or population size in a geographical region to aid comparisons. Mortality below the expected levels is called “avoided mortality”, whereas the mortality above the expected levels is known as “excess deaths” [3]. Assessing the direct and indirect effects of the COVID-19 pandemic on overall mortality requires the measurement of excess deaths since most excess deaths are likely attributable to COVID-19 [3,8].

Worldwide, Peru is the country with the highest number of COVID-19 deaths per 100,000 population [9]. In 2020, 93,851 COVID-19 deaths were registered in the country, and by 26 December 2021, the total had reached 202,524 deaths [10]. Because of the Peruvian national healthcare system’s limited capacity, the collapse of health services in the first wave, limited number of intensive care unit (ICU) beds, lack of oxygen [11,12], and the high COVID-19 death rate, excess all-cause mortality is likely attributable to COVID-19. Therefore, our objective was to compare the all-cause excess mortality with the COVID-19 mortality in 25 Peruvian regions to determine whether most of the excess deaths in 2020 were attributable to COVID-19. These findings could be used to determine the indirect impact of the COVID-19 pandemic on the overall mortality rate in Peru.

## 2. Materials and Methods

This cross-sectional, geographical time-series study was performed according to the Strengthening the Reporting of Observational Studies in Epidemiology (STROBE) reporting guidelines [13]. We retrieved disaggregated region-level data on confirmed COVID-19 deaths and all-cause mortality, as of 31 December 2020, from the Sistema Informatico Nacional de Defunciones (SINADEF) at the Ministry of Health of Peru [14,15]. We used death registers from 1 January through to 31 December 2020 (1–52 epidemiological weeks) and from the preceding 3 years (2017–2019) [14,15]. Data regarding the populations of Peruvian regions were obtained from the projections of the Instituto Nacional de Estadística e Informática (INEI) [16]. Confounding factors included the mean age, gender balance, and number of ICU beds for each region (from 2020). The mean age and gender balance in the regions were obtained from INEI (from 2016 to 2020) [16]. The number of ICU beds was obtained from the Superintendencia Nacional de Salud, Peru (SUSALUD) via App. F500.2 [17].

### Statistical Analysis

The average numbers of all-cause deaths for the years 2017–2019 were used to estimate expected deaths in 2020 [8,18]. Observed deaths were the deaths reported from 1 January through to 31 December 2020. Excess all-cause deaths during the pandemic period were estimated as the difference between observed deaths and expected deaths in 2020 [8,18]. We calculated the excess all-cause mortality rate and COVID-19 mortality rate by region per 100,000 population. Excess deaths attributable to COVID-19 were calculated (%) by dividing COVID-19 deaths per 100,000 by excess deaths per 100,000 population. Spearman’s test and a linear regression model were used to estimate the correlation between excess all-cause mortality rate and COVID-19 mortality rate per 100,000 population. Multiple regression analysis was also used for confounding factors (mean age in the region, gender balance, and number of ICU beds). Values of *p* < 0.05 were considered significant. Results were displayed using a scatterplot. All analyses were performed using StataSE 16.0 for Windows.

This descriptive study was based on public-use datasets. Therefore, it was exempt from Institutional Review Board review and approval, and no informed consent was required.

## 3. Results

All Peruvian regions experienced an all-cause mortality excess in 2020, compared with expected deaths (determined from the mean between 2017 and 2019). Excess all-cause death rates varied widely among regions (range: 115.1 to 519.8 per 100,000 population). The ratio of observed to expected all-cause deaths ranged between 1.5 and 2.8. COVID-19 death rates ranged between 83.8 and 464.6 per 100,000 population, and excess deaths (%) ranged between 48.8 and 108.3% (Table 1). In the general population of Peru, the excess all-cause mortality exceeded COVID-19 mortality (371.9 vs. 287.7 population, respectively). There were variations in excess all-cause mortality and COVID-19 mortality by region. The highest excess all-cause mortality per 100,000 habitants was reported in the Callao region, followed by Lima, Moquegua, and Piura regions. The highest COVID-19 mortality rates per 100,000 habitants were reported in Moquegua, Lima, Ica, and Lambayeque. In 19 Peruvian regions, the ratio of excess all-cause deaths to COVID-19 deaths was almost 1 (Table 1). In six Peruvian regions, there was a gap between the all-cause excess mortality and COVID-19 mortality, e.g., in the Apurimac, Huancavelica, and Pasco regions, the ratio of excess all-cause deaths to COVID-19 deaths was 2.0, while in the Ayacucho, Cajamarca, and Puno regions it was almost 2.0 (Table 1).

There was a correlation between the all-cause excess mortality rate and the COVID-19 mortality rate (r = 0.90; *p* = 0.00001; y = 0.8729x + 90.808; R^2^ = 0.84) (Figure 1). Adjusted for confounding factors (mean age in the region, gender balance, and number of ICU beds), the all-cause excess mortality rate was correlated with the COVID-19 mortality rate (β = 0.921; *p* = 0.0001) (Table 2). The model was statistically significant (F (4,20) = 37.46, *p* = 0.00001, Adj. R^2^ = 0.882).

## 4. Discussion

Worldwide, Peru is the country with the highest number of COVID-19 deaths per 100,000 population [9]. This has caused an excess of all-cause mortality [4] and compared with other countries, in 2020 Peru experienced the largest excess mortality among 103 countries studied [1]. This excess all-cause mortality recorded in 2020 is clearly related to the health crisis caused by the COVID-19 pandemic. This was shown by the ratio of excess all-cause deaths to COVID-19 deaths and the adjusted analysis of death rates reported in the same period in most Peruvian regions. In addition, we found that COVID-19 mortality as excess all-causes mortality varied widely between Peruvian regions. Similar to Peru, some Brazilian states, Iran, and Belgium reported an excess all-cause mortality during the first wave proportional to the number of people who died of COVID-19 in the same period [19,20,21].

Excess mortality during the COVID-19 pandemic can be the sum of distinct factors. These factors include: (1) deaths directly caused by COVID-19 infection, (2) medical system collapse due to COVID-19 pandemic, (3) excess deaths from other natural causes, (4) unnatural causes, or (5) extreme events [1]. Compared with other factors, most excess deaths are likely attributable to COVID-19 infection [1,8]. In that setting, the literature has described some factors that directly or indirectly impact COVID-19 mortality in Peru [1,11,12], and therefore on excess all-cause mortality (Figure 2).

The factors that may have contributed to the causal relationship between all-cause excess mortality and COVID-19 mortality in Peru and the large differences in excess mortality from one region of Peru to another could have several explanations. First, mortality rates depend on social factors such as demographic and socio-economic characteristics, including age, population structure, population size, lifestyles, obesity prevalence, ethnicity, and the mobility of populations across between regions, as was observed in several countries, including Peru [2,12,22,23]. Second, mortality rates also depend on the probability of being infected, prevalence and incidence rates, and mortality among the infected population, since worldwide, Peru was the country with the highest number of COVID-19 deaths per 100,000 population [9]. Third, because of the Peruvian national healthcare system’s limited capacity, excess all-cause mortality may be a more comprehensive and robust indicator than COVID-19 mortality. Thus, the collapse of the health services in Peru, the fragmented health system, the limited number of ICU beds, and lack of oxygen during the first wave may have also contributed directly or indirectly to the increased relationship between the death rate due to COVID-19 and the excess all-cause mortality [11,12]. A recent study found a gap between excess mortality and COVID-19 deaths in 67 countries, including Peru [24]. Their findings revealed that the countries where COVID-19 mortality exceeded excess all-cause mortality had an extremely high testing capacity and effective response measures against the COVID-19 pandemic. In contrast, the excess all-cause mortality exceeded COVID-19 mortality in the general population of Peru, because there was a low rate of RT-PCR testing for COVID-19 in Peru in 2020; therefore, most of the cases were diagnosed using rapid tests, the sensitivity limits of which are low compared with molecular tests. Because of this, it is possible that some of the deaths recorded as other causes might have been due to COVID-19; consequently, the excess all-cause death rate increased and was greater than the COVID-19 mortality. Fourth, individual factors such as comorbidities and genetic and immunological factors also directly or indirectly impact COVID-19 mortality [25], and therefore the excess all-cause mortality can be seen as an indirect consequence of COVID-19 mortality. In addition, the COVID-19 mortality in Peru was highest in men 60 years of age or older [26], and with co-morbidities [27]. This could have caused difficulty in identifying the basic cause of death, and therefore an underreporting of COVID-19 deaths that increased the excess all-cause mortality. Finally, COVID-19 is a new disease and physicians have limited experience in certifying these deaths, which may have resulted in deaths being underreported in the first months of 2020.

During the COVID-19 pandemic, it is likely that deaths from non-COVID-19 causes may also have increased due to the medical system being overloaded. However, to date, in Peru, there are no studies on excess mortality from non-COVID-19 causes. The restrictive measures adopted in Peru to control the COVID-19 pandemic in the first months of 2020 (COVID-19 lockdown) also could have caused changes in mortality rates due to external causes such as injuries or accidents, as was reported previously in England [28]; however, this did not happen in Peru, since in 2020 there was a decrease in the mortality rates by homicides, suicides, and traffic accidents during the COVID-19 lockdown [29].

The main limitation of our study was the method used to estimate excess mortality. In the literature, the methods for estimating excess mortality vary from simple estimates to modeled studies, making it difficult to compare results across studies. Another limitation in this observational study is the retrospective design; we used several different information sources (SINADEF, INEI, and SUSALUD) which may have resulted in a possible bias. Despite these limitations, the strengths of this study include: (1) the used method allows for transparency and reproducibility of the findings; (2) the simplicity in the analysis of excess mortality (all-cause and COVID-19 mortality) allows for the opportunity to use the findings in epidemiological surveillance and their interpretation by the health authorities; (3) the large number of deaths included for estimating the excess all-cause mortality and COVID-19 mortality and the multiple regression analyses for confounding factors; and (4) the findings add further evidence for policymakers in Peru.

## 5. Conclusions

This study provides the first causal relationship analysis between excess all-cause mortality and COVID-19 mortality for 2020 across 25 Peruvian regions, adjusted for confounding factors. Our findings suggest that most of the excess deaths in Peru in 2020 were related to COVID-19. Therefore, our findings could be used to explain the indirect impact of the COVID-19 pandemic on the overall mortality rate up to the point where vaccination against SARS-CoV-2 started to become available in Peru.

## Figures and Tables

**Figure 1 tropicalmed-07-00044-f001:**
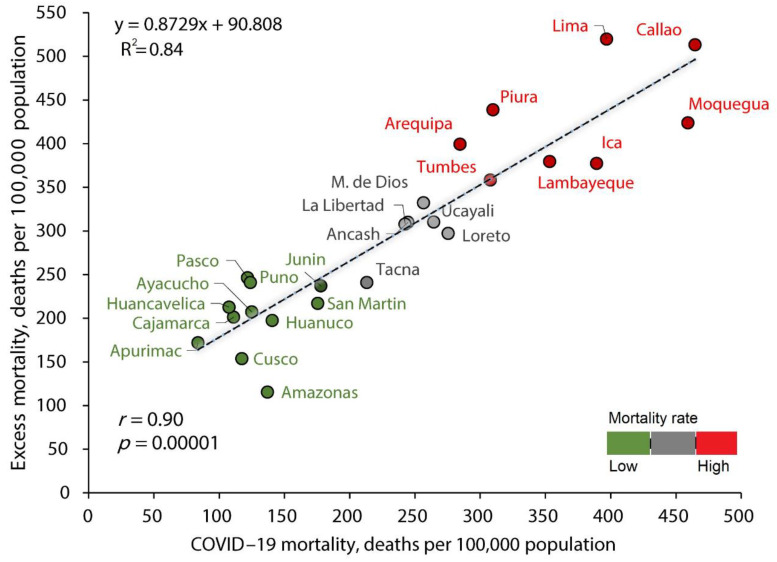
Correlation between the all-cause excess mortality and COVID-19 mortality in Peru.

**Figure 2 tropicalmed-07-00044-f002:**
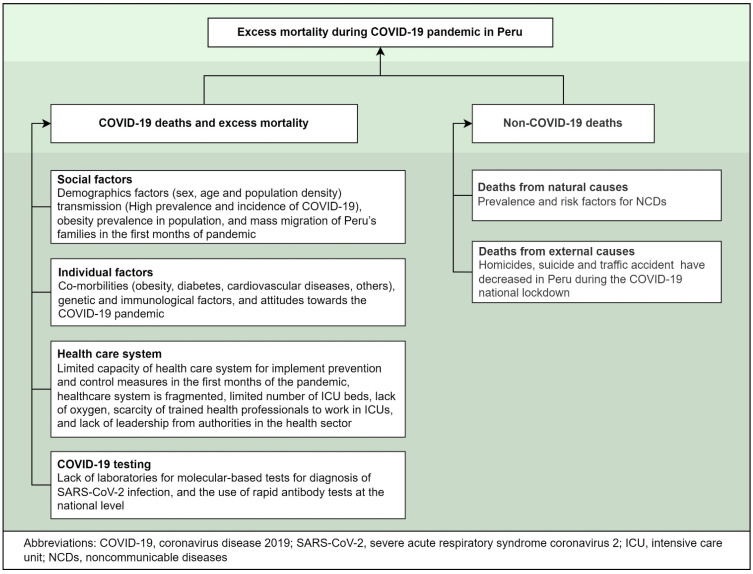
Factors that directly or indirectly impact excess all-cause mortality and COVID-19 mortality [1,8,11,12].

**Table 1 tropicalmed-07-00044-t001:** Excess all-cause deaths and COVID-19 mortality, 1 January to 31 December 2020, 25 Peruvian regions.

Region	Observed Deaths in 2020 [15]	Expected Deaths in 2020 [15]	Ratio of Observed to Expected	Population in 2020 [16]	Excess Deaths	Excess Deaths per 100,000	Total Deaths COVID-19 [14]	COVID-19 Deaths per 100,000	Excess Deaths Attributable to COVID-19, % ^a^	Ratio of Excess Deaths to COVID-19 Deaths	Mean Age (Years) in the Regions in 2020 [16]	Gender Balance (Men/Women) in 2020 [16]	No. of ICU Beds ^b^
Amazonas	1309	818	1.6	426,806	491	115.1	585	137.1	119.1	0.8	30.95	1.1	10
Ancash	8800	5143	1.7	1,180,638	3657	309.7	2889	244.7	79.0	1.3	32.36	1.0	18
Apurimac	2242	1502	1.5	430,736	740	171.7	361	83.8	48.8	2.0	29.99	1.1	10
Arequipa	11,634	5655	2.1	1,497,438	5979	399.3	4260	284.5	71.2	1.4	32.45	1.0	22
Ayacucho	3097	1713	1.8	668,213	1384	207.1	836	125.1	60.4	1.7	30.04	1.0	15
Cajamarca	6041	3116	1.9	1,453,711	2925	201.2	1615	111.1	55.2	1.8	30.71	1.0	18
Callao	10,124	4327	2.3	1,129,854	5797	513.1	5249	464.6	90.5	1.1	32.36	0.9	16
Cusco	7857	5775	1.4	1,357,075	2082	153.4	1594	117.5	76.5	1.3	30.34	1.0	14
Huancavelica	2457	1680	1.5	365,317	777	212.6	393	107.6	50.6	2.0	30.28	1.0	12
Huanuco	4268	2768	1.5	760,267	1500	197.3	1070	140.7	71.3	1.4	30.01	1.0	25
Ica	7785	4105	1.9	975,182	3680	377.4	3796	389.3	103.2	1.0	30.93	1.0	26
Junin	8573	5347	1.6	1,361,467	3226	236.9	2423	178.0	75.1	1.3	31.18	1.0	42
La Libertad	13,800	7594	1.8	2,016,771	6206	307.7	4891	242.5	78.8	1.3	31.55	1.0	26
Lambayeque	8421	3447	2.4	1,310,785	4974	379.5	4631	353.3	93.1	1.1	32.24	0.9	22
Lima	87,139	31,889	2.7	10,628,470	55,250	519.8	42,182	396.9	76.3	1.3	33.05	0.9	367
Loreto	5102	2050	2.5	1,027,559	3052	297.0	2832	275.6	92.8	1.1	28.50	1.1	2
Madre de Dios	1089	512	2.1	173,811	577	332.2	446	256.6	77.3	1.3	27.48	1.4	7
Moquegua	1574	757	2.1	192,740	817	423.9	885	459.2	108.3	0.9	32.85	1.2	6
Pasco	1234	564	2.2	271,904	670	246.3	331	121.7	49.4	2.0	30.12	1.1	6
Piura	13,974	4992	2.8	2,047,954	8982	438.6	6345	309.8	70.6	1.4	30.42	1.0	26
Puno	8173	5192	1.6	1,237,997	2981	240.8	1535	124.0	51.5	1.9	29.72	1.0	14
San Martin	4361	2409	1.8	899,648	1952	217.0	1579	175.5	80.9	1.2	29.89	1.1	14
Tacna	2167	1274	1.7	370,974	893	240.8	791	213.2	88.5	1.1	31.91	1.0	20
Tumbes	1796	895	2.0	251,521	901	358.4	774	307.7	85.9	1.2	29.91	1.2	8
Ucayali	3533	1706	2.1	589,110	1827	310.1	1558	264.5	85.3	1.2	28.00	1.1	12
Peru	226,550	105,229	2.2	32,625,948	121,321	371.9	93,851	287.7	77.4	1.3	NA	1.0	758

^a^ Excess deaths attributable to COVID-19 calculated (%) by dividing COVID-19 deaths per 100,000 by excess deaths per 100,000 population. ^b^ SICOVID App. F500.2, SUSALUD (accessed on 30 September 2020). COVID-19, coronavirus disease 2019; NA, not aplicable; ICU, intensive care unit.

**Table 2 tropicalmed-07-00044-t002:** Multiple regression analysis of mortality all-cause excess rate and COVID-19 mortality rate adjusted.

Variable	Coef.	SE	*Beta*	*t*	*p*-Value
Mortality all-cause excess and COVID-19 mortality					
COVID-19 mortality rate	0.875	0.088	0.921	9.89	0.0001
Mean age (years) in the region	−9.500	8.673	−0.126	−1.10	0.286
Gender balance	−138.9	114.4	−0.128	−1.21	0.239
Number of ICU beds	0.257	0.132	0.167	1.95	0.065

COVID-19, coronavirus disease 2019; SE, standard error; ICU, intensive care unit.

## Data Availability

The data presented in this study are publicly available at: COVID-19 deaths. National System of Deaths (SINADEF). Available online: https://www.datosabiertos.gob.pe/dataset/fallecidos-por-covid-19-ministerio-de-salud-minsa (accessed on 26 December 2021); Deaths. National System of Deaths (SINADEF). Available online: https://www.datosabiertos.gob.pe/dataset/informaci%C3%B3n-de-fallecidos-del-sistema-inform%C3%A1tico-nacional-de-defunciones-sinadef-ministerio (accessed on 26 December 2021); INEI, Peruvian population: https://www.inei.gob.pe/estadisticas/indice-tematico/population-estimates-and-projections/ (accessed on 26 December 2021); Daily Report on Form F500.2, app. for centralized management of the availability of Hospitalization and ICU beds at the national level and of all subsystems (Application F500.2): http://portal.susalud.gob.pe/seguimiento-del-registro-de-camas-f500-2/ (accessed on 26 December 2021).

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
