# Peer review of "Analysis of Excess All-Cause Mortality and COVID-19 Mortality in Peru: Observational Study"

_tropicalmed, 2022, doi:10.3390/tropicalmed7030044_

Round 1
Reviewer 1 Report
The article discusses a topical issue in an international context.
The purpose and method of work are well defined.
It is based on public data, statistically processed in a relevant way.
It is very concise, without digressions, but also without more advanced scientific deepening of the interpretation, beyond the statistical processing and some correlations with the results of other research.
The main weakness, in my opinion, is the lack of a causal interpretation of the large differences in excess mortality (due to general causes and due to Covid 19) from one region of Peru to another. I consider that this analysis needs to be done in order to strengthen the scientific validity of the study, but especially the practical utility, for the decision makers at national and regional level, of which you as authors reiterate it.
Author Response
The article discusses a topical issue in an international context.
The purpose and method of work are well defined.
It is based on public data, statistically processed in a relevant way.
We thank the Reviewer for your comments and constructive criticism, we believe that the quality of our manuscript has been significantly improved. We have revised our paper in a point-by-point manner. Modifications are in yellow text.
Comment 1: It is very concise, without digressions, but also without more advanced scientific deepening of the interpretation, beyond the statistical processing and some correlations with the results of other research.
Response 1: Thank you for your comment. We expanded the Discussion section and included Figure 1. Factors that directly or indirectly impacting on excess all-cause mortality and COVID-19 mortality [1,8,11,12].
Comment 2: The main weakness, in my opinion, is the lack of a causal interpretation of the large differences in excess mortality from one region of Peru to another. I consider that this analysis needs to be done in order to strengthen the scientific validity of the study, but especially the practical utility, for the decision makers at national and regional level, of which you as authors reiterate it.
Response 2: Thank you for your comment. This is a country-level study. We reviewed the literature and explained all the factors that may have contributed to the causal relationship between all-cause excess mortality and COVID-19 mortality in Peru, and the large differences in excess mor-tality from one region of Peru. We expanded the Discussion section and included Figure 1. Factors that directly or indirectly impacting on excess all-cause mortality and COVID-19 mortality [1,8,11,12].
Reviewer 2 Report
This article is clear, well-written and appears to describe relevant findings to the field. I have only minor comments/recommendations for this manuscript.
- In Figure one, please indicate the reasoning behind color coding the region dots. I believe it's population based, but test in the figure legend would be nice to confirm where cutoffs were established.
- Citation typo in line 123 (I believe you are intending to cite citation 3, but it's just a random 3 right now)
- This is not required, but I would say both the Introduction and Results section are a bit short. The introduction is almost a perfect repeat of the abstract. Also, Table 1 presents quite a bit of data that I think the authors could spend more time unpacking. I'd like to see at least a few more sentences, if not another paragraph in both.
Author Response
We thank the Reviewer for your comments and constructive criticism, we believe that the quality of our manuscript has been significantly improved. We have revised our paper in a point-by-point manner. Modifications are in yellow text.
Comment 1: In Figure one, please indicate the reasoning behind color coding the region dots. I believe it's population based, but test in the figure legend would be nice to confirm where cutoffs were established.
Response 1: Thank you for your comment. The sentence Figure 1 was corrected
Comment 2: Citation typo in line 123 (I believe you are intending to cite citation 3, but it's just a random 3 right now)
Response 2: The error was corrected
Comment 3: This is not required, but I would say both the Introduction and Results section are a bit short.
Response 4: We have included a paragraph in the Results section (lines 93-97).
Comment 4: The introduction is almost a perfect repeat of the abstract.
Response 4: We have included a paragraph in the Introducción section (lines 31-43).
Comment 5: Also, Table 1 presents quite a bit of data that I think the authors could spend more time unpacking. I'd like to see at least a few more sentences, if not another paragraph in both.
Response 5: We have included a paragraph in the Results section (lines 93-97).
Reviewer 3 Report
The manuscript submitted for review is interesting and valid.
TITLE
Improve its clarity and loudness.
ABSTRACT
A definition of excess death is recommended.
Avoid the translation of organizations: INEI
INTRODUCTION
Define excess deaths.
Contextualize COVID worldwide and in Peru. I suggest using this reference as a base bibliography. Tadj A, Lahbib-Seddiki SM. Our Overall Current Knowledge of Covid 19: An Overview. Microbes Infect Chemother. 2021; 1: e1262. https://doi.org/10.54034/mic.e1262
METHODS
Reference 11 in square brackets.
Indicate from what year is the number of ICU beds available as well as the sex and average age.
Make a small ethical statement that there is no sensitive information in the data and the codes of ethics were complied with.
DISCUSSION
Mention, as possible causes of excess deaths, self-medication and the use of drugs without proven efficacy by the population.
Author Response
We thank the Reviewer for your comments and constructive criticism, we believe that the quality of our manuscript has been significantly improved. We have revised our paper in a point-by-point manner. Modifications are in yellow text.
The manuscript submitted for review is interesting and valid.
Comment 1: TITLE. Improve its clarity and loudness.
Response 1: Thank you for your comment. We consider the title "Analysis of Excess All-cause Mortality and COVID-19 Mortality in Peru: Observational Study"
Comment 2: ABSTRACT. A definition of excess death is recommended.
Response 2: Thank you for your comment. We include a definition of excess mortality.
Comment 3: Avoid the translation of organizations: INEI.
Response 3: Thank you for your comment. The sentence was corrected.
Comment 4: INTRODUCTION. Define excess deaths.
Response 4: Thank you for your comment. We include a definition of excess mortality.
Comment 5: Contextualize COVID worldwide and in Peru. I suggest using this reference as a base bibliography. Tadj A, Lahbib-Seddiki SM. Our Overall Current Knowledge of Covid 19: An Overview. Microbes Infect Chemother. 2021; 1: e1262. https://doi.org/10.54034/mic.e1262
Response 4: Thank you for your comment. In the manuscript Tadj et al. we found no information on excess mortality. Instead we include the refs. 5-8, 12, 22-24, 26
Comment 6: Reference 11 in square brackets.
Response 6: Thank you for your comment. This error was corrected.
Comment 7: Indicate from what year is the number of ICU beds available as well as the sex and average age.
Response 7: Thank you for your comment. We include the years (lines 63-65).
Comment 8: Make a small ethical statement that there is no sensitive information in the data and the codes of ethics were complied with.
Response 8: Thank you for your comment. We include a sentence on ethical statement (lines 83-85).
Comment 9: Mention, as possible causes of excess deaths, self-medication and the use of drugs without proven efficacy by the population.
Response 9: Thank you for your comment. We have reviewed the literature, but found no studies on possible causes of excess deaths, self-medication and the use of drugs without proven efficacy by the population to support our findings (excess mortality). Instead we expanded the Discussion section and included Figure 1. Factors that directly or indirectly impacting on excess all-cause mortality and COVID-19 mortality [1,8,11,12].